# Oligometastatic Prostate Adenocarcinoma. Clinical-Pathologic Study of a Histologically Under-Recognized Prostate Cancer

**DOI:** 10.3390/jpm10040265

**Published:** 2020-12-04

**Authors:** Claudia Manini, Alba González, David Büchser, Jorge García-Olaverri, Arantza Urresola, Ana Ezquerro, Iratxe Fernández, Roberto Llarena, Iñaki Zabalza, Rafael Pulido, Arkaitz Carracedo, Alfonso Gómez-Iturriaga, José I. López

**Affiliations:** 1Department of Pathology, San Giovanni Bosco Hospital, 10154 Turin, Italy; claudiamaninicm@gmail.com; 2Department of Radiation Oncology, Cruces University Hospital, 48903 Barakaldo, Spain; albagl.91@gmail.com (A.G.); david.buechsergarcia@osakidetza.eus (D.B.); 3Department of Urology, Cruces University Hospital, 48903 Barakaldo, Spain; jorge.garciaolaverrirodriguez@osakidetza.eus (J.G.-O.); roberto.llarenaibarguren@osakidetza.eus (R.L.); 4Department of Radiodiagnostics, Cruces University Hospital, 48903 Barakaldo, Spain; aranzazu.urresolaolabarrieta@osakidetza.eus (A.U.); anaisabel.ezquerroimas@osakidetza.eus (A.E.); 5Department of Nuclear Medicine, Cruces University Hospital, 48903 Barakaldo, Spain; iratxe.fernandez@osakidetza.eus; 6Department of Pathology, Galdakao Hospital, 48960 Galdakao, Spain; inaki.zabalzaestevez@osakidetza.eus; 7Biocruces-Bizkaia Health Research Institute, 48903 Barakaldo, Spain; rpulidomurillo@gmail.com; 8Ikerbasque, The Basque Foundation for Science, 48009 Bilbao, Spain; acarracedo@cicbiogune.es; 9Department of Cell Biology and Histology, Faculty of Medicine and Nursing, University of the Basque Country (UPV/EHU), 48940 Leioa, Spain; 10Center for Cooperative Research in Biosciences, Basque Research and Technology Alliance (BRTA), 48160 Derio, Spain; 11CIBERONC, 28046 Madrid, Spain; 12Department of Pathology, Cruces University Hospital, 48903 Barakaldo, Spain

**Keywords:** prostate cancer, oligometastatic disease, histopathology, inflammation, atrophy, gleason index, pathologic staging, immunohistochemistry, prognosis

## Abstract

The clinical parameters and the histological and immunohistochemical findings of a prospective protocolized series of 27 prostate carcinoma patients with oligometastatic disease followed homogeneously were analyzed. Lymph nodes (81.5%) and bones (18.5%) were the only metastatic sites. Local control after metastatic directed treatment was achieved in 22 (81.5%) patients. A total of 8 (29.6%) patients developed castration-resistant prostate cancer. Seventeen (63%) patients presented with non-organ confined disease. The Gleason index 8–10 was the most frequently observed (12 cases, 44.4%) combined grade. Positive immunostainings were detected with androgen receptor (100%), PGP 9.5 (74%), ERG (40.7%), chromogranin A (29.6%), and synaptophysin (18.5%) antibodies. The Ki-67 index value > 5% was observed in 15% of the cases. L1CAM immunostaining was negative in all cases. Fisher exact test showed that successful local control of metastases was associated to mild inflammation, organ confined disease, Ki-67 index < 5%, and Gleason index 3 + 3. A castration resistant status was associated with severe inflammation, atrophy, a Gleason index higher than 3 + 3, Ki-67 index ≥ 5%, and positive PGP 9.5, chromogranin A, and synaptophysin immunostainings. In conclusion, oligometastatic prostate adenocarcinoma does not have a specific clinical-pathologic profile. However, some histologic and immunohistochemical parameters of routine use may help with making therapeutic decisions.

## 1. Introduction

Prostate adenocarcinoma (PCa) is the most common neoplasm and a leading cause of mortality in the male population in Western countries, with more than 191,000 new cases and 33,000 deaths expected in the USA in 2020 [1]. The biology of this tumor is complex and sets forth very different clinical contexts that are not always easy to manage in practice.

A high percentage of PCa are detected in an organ-confined stage, and radical surgery or radiotherapy is the recommended treatment [2]. In this group of patients, however, the definition and management of the so-called clinically insignificant disease remains controversial [3], as well as the role of active surveillance [4,5] and focal therapy [6] as alternative options.

On the other side of the spectrum, a significant subset of patients presents with a disseminated poly-metastatic status early in the clinical evolution, even at diagnosis. PCa in these patients encompasses high levels of intratumor heterogeneity and a wide spectrum of genomic alterations [7], with androgen-deprivation therapy as the main alternative with or without synchronous or metachronous chemotherapy or second-line antiandrogen regimes [8].

In between these two clinical contexts, a group of PCa patients is characterized by developing what it is called an oligometastatic status (OPCa), that is, a clinical situation in which a limited number of metastases, usually between one and five, appear during the course of the disease [9]. Since OPCa still has not been morphologically, immunohistochemically, or genomically characterized, this group of patients remains stratified simply under a clinical concept with unpredictable prognosis and treatment that has not yet been established. In this particular context, there are no specific treatment recommendations for limited metastatic disease. Different treatment approaches are currently used, most of them focusing on local ablative modalities using surgery or radiation [10].

This prospective study aims to describe the clinical and pathologic characteristics of a series of patients with OPCa initially treated with radical prostatectomy. The histological and immunohistochemical features of 27 primary tumors which subsequently developed an oligometastatic status have been evaluated.

## 2. Materials and Methods

This study was conducted under approval of the Cruces University Hospital Ethical Committee (PI2019059) and is compliant with the principles outlined in the Declaration of Helsinki. All patients agreed and signed a consent form.

A total of 27 patients initially treated with radical prostatectomy have been collected at the Department of Radiotherapy Oncology, Cruces University Hospital, Barakaldo, Spain. All patients developed an asymptomatic relapse with ≤5 metachronous metastases after surgery +/− post-operative radiotherapy. The diagnosis and treatment protocol has been described elsewhere [11]. Briefly, oligometastases were diagnosed based on Choline PET/CT scans and were treated with Stereotactic Ablative Radiotherapy (SABR) and/or Volumetric Modulated Arc Therapy (VMAT), and/or Intensity-Modulated Radiation Therapy (IMRT) at our institution between December 2012 and February 2017. Data for these patients were prospectively collected and analyzed.

Following the currently accepted criteria [9], patients were included in the study when they developed up to five metastases during their clinical follow-up, either in the same or different organs. Demographic data and the main clinical findings of all patients were annotated. The indication of metastases directed treatment (MDT) was determined based on the following criteria: a previously treated primary tumor and an apparent complete local response, more than 2 years from diagnosis of the primary tumor to the metastatic recurrence, availability of Ch-PET/CT parameters at positive lymph node (LN) or bone metastases, and a maximum of 5 metastatic sites and LN with a diameter of 3 cm or less.

The nadir plus 2 ng/mL definition was used to define biochemical relapse, a clinical disease-free survival (cDFS) event was defined as clinical evidence of disease by any clinical, pathological, or radiological method. Finally, the time to castrate-resistant disease was also estimated.

Patients were followed-up with clinical examinations at 1, 3, and 6 months in all cases, and thereafter every 1 to 3 months. Prostate-specific antigen (PSA) measurements were scheduled 3–6-monthly during the first year and 6-monthly thereafter. Reassessment with Ch-PET/CT imaging was performed in case of 3 rising PSA values after initial response and a PSA value higher than 2 ng/mL or if clinically indicated to exclude local or distant metastatic progression. In the case of an oligometastatic recurrence outside the previous SABR/IMRT field, retreatment with SABR/IMRT was offered.

Representative samples of the primary tumor fixed in formalin and embedded in paraffin blocks following routine protocols were analyzed by three pathologists (CM, IZ, JIL) which blindly evaluated the same pathologic criteria: histologic type, Gleason index, 2014 ISUP prognostic grouping [12], UICC/AJCC staging system (8th edition) [13], the presence of intratumor inflammatory cells [quantified as mild (occasional lymphoid aggregates), moderate (one lymphoid aggregate every two high power fields), and intense (one or more lymphoid aggregates per high power field)], and the presence of atrophic changes and stromal reaction in the same histological slides stained with hematoxylin-eosin.

Immunomarkers related with tumor aggressiveness (neurogenesis, proliferative index, neuroendocrine differentiation, and angiogenesis) have been tested in an attempt to delineate, coupled with classical histologic features, any specific profile of this under-recognized clinical presentation of PCa. For such a purpose, Androgen receptor (SP107, Ventana, 760-4605, ready to use, nuclear staining), ERG (EPR3864, Ventana, 790-4576, ready to use, nuclear and cytoplasmic staining), PGP 9.5 (Ventana, 760-4434, ready to use, cytoplasmic staining), Ki67 (30-9, Ventana, 790-4286, ready to use, nuclear staining), chromogranin A (LK2H10, Ventana, 760-2519, ready to use, cytoplasmic staining), synaptophysin (MRQ-40, Ventana, 760-4595, ready to use, cytoplasmic staining), and L1CAM (UJ127.11, Sigma, L4543, dilution 1:100) staining were performed in automated immunostainers (BenchMark Ultra, Ventana Medical Systems) following routine methods. Tris-EDTA was used for antigen retrieval. Negative controls were slides not exposed to the primary antibody, and these were incubated in PBS and then processed under the same conditions as the test slides.

Descriptive statistics were used to summarize patient, tumor, and treatment characteristics. Chi-square and Fisher exact test was used to analyze the relation between histopathological features and clinical outcomes.

## 3. Results

Twenty-seven patients treated with radical prostatectomy and prospectively followed were included in the study. The average follow up was 54 months (range, 26–70). The clinical characteristics are presented in Table 1.

At the time of biochemical relapse after radical prostatectomy, the average age of the patients was 67 years (range, 56–78). Serum PSA levels oscillated between 1.03 and 12.3 ng/mL (average, 3.41 ng/mL). The average PSA doubling time was 4.5 months (range, 1–21). Metastases were located in LN (22/27 cases, 81.5%) and bones (5/27 cases, 18.5%).

Local control after the MDT (SABR or VMAT) was successful in 22 cases (81.5%). At relapse after initial MDT, 11 out of 27 patients (40.7%) developed oligometastatic disease and received subsequent MDT. Of these, 3 patients developed OPCa and were also treated locally with MDT. The treatment in all newly appearing metastases was always the same (SABR or VMAT).

Twenty-two patients (81.5%) developed biochemical failure after the first MDT. The average of biochemical failure free survival was 21 months (95%, CI 52.20–57.79). Five patients (18.5%) developed a local relapse of the treated metastases. The average clinical progression free was 55 months (95%, CI 10.82–31.17). Nine out of 27 (33.3%) patients developed a castration-resistant status during the follow-up.

### 3.1. Pathological Data

The results are shown in Table 2. Pathological staging distribution showed 17 (63%) non-organ confined (pT3a/b) versus 10 (37%) confined (pT2a/b/c) cases. Twenty-five cases (92.5%) were conventional acinar adenocarcinomas and 2 (7.5%) were duct adenocarcinomas. Intratumor lymphoid infiltrates were mild in 8 cases (29.6%), moderate in 10 cases (37%), and intense in 9 cases (33.3%). Atrophic changes were detected in the adjacent benign prostate tissue in 18 cases (66.6%) and tumor stromal fibroblastic reaction in 15 cases (55.5%). The Gleason score (GS) distribution was as follows: GS = 6, 3 cases (11.1%); GS = 3 + 4, 9 cases (33.3%); GS = 4 + 3, 3 cases (11.1%); and GS = 8 to 10, 12 cases (44.4%).

Immunohistochemistry displayed positivity for the androgen receptor in 27/27 cases (100%), ERG in 11/27 cases (40.7%), PGP 9.5 in 20/27 cases (74%), chromogranin A in 8/27 cases (29.6%), and synaptophysin in 5/27 cases (18.5%). Ki67 expression was ≤ 5% in 23/27 cases (85.1%). L1CAM was negative in all cases.

### 3.2. Statistical Analysis

The results of the Chi-square (Fisher exact test) are summarized in Table 3. They showed that local control after MDT was associated to non-organ confined tumor category, low Gleason index, stromal reaction, mild inflammation, atrophy, Ki67 > 5% and ≥ 5%, ERG, PGP 9.5, CRG, and SYN immunostainings. Non-organ confined disease at diagnosis was associated to mild inflammation, Ki-67 index > 5% and ≥ 5%, chromogranin A and synaptophysin expression. The development of castration-resistant tumors, mild inflammation, atrophy, low Gleason index, CRG, SYN, PGP 9.5, Ki-67 index > 5% and ≥ 5% were shown to be dependent variables.

## 4. Discussion

OPCa is a tumor category strictly defined by clinical-radiological criteria [9,14,15]. Most series classify as OPCa those cases that develop less than 3–5 metastases during the follow up of the disease [9]. These cases are considered as an intermediate clinical status between localized, organ-confined, and polymetastatic diseases. Either active surveillance [16] or metastasis-directed therapies [17] may be good options to delay systemic treatments in these patients. Hence, the interest generated to identify such cases early at the pre-metastatic phase of the disease. Up to now, however, no distinctive features have been detected at the microscopic level. Oligometastases are usually asymptomatic, so their detection needs close follow up protocols and precise imaging methodologies due to the potential benefit of local therapies in these patients [18].

Little is known about the biological reasons for which a tumor may develop either none, few, or many metastases along its evolution. So far, there is no specific histological, immunohistochemical, or molecular feature linked to this clinical behavior in PCa. Metastasis development is a complex stepwise process of adaptation in which tumor cells and the local microenvironment interplay [19]. Speaking in quantitative terms, metastasis is not an efficient process. It has been calculated that only around 0.02% of circulating tumor cells succeed developing a distant metastasis [20]. For such a purpose, a subset of tumor cells in the primary tumor acquire a specific tropism that includes the secretion of cytokines and growth factors to create the successful local conditions in the target organ for posterior colonization following the Paget’s hypothesis [21]. These favorable environmental conditions are site-specific and conform what is called the pre-metastatic niche [22]. Latterly, once arrived in the niche, the metastatic seed needs to initiate immune evasion processes to survive to the local innate immunity transforming macrophages, NK cells, and other elements in allied conspirators. Then arrives an unprecise period of dormancy followed by epigenetic reprogramming processes that will allow PCa metastatic cells to survive and escape therapies, particularly the androgen receptor signaling blockade, thus converting the neoplasm in a castration-resistant cancer, the common final pathway in the clinical evolution of PCa.

Chronic inflammation, atrophy, and stromal fibroblastic induction [23,24] have been classically associated with prostate cancer initiation and progression. The current findings show that the microbiome associated with chronic inflammation and atrophy allows the development of a specific microenvironment promoting prostate carcinogenesis [24]. Stromal matrix remodeling replaces the prostate fibromuscular stroma and evolves during prostate cancer evolution through transforming growth factor β1 regulation [23]. Besides, a very recent study demonstrates that collagen production and accumulation are driven by the activation of PI3K in the prostate epithelium [25]. In this sense, prominent chronic inflammation has been detected in the present series in 33.3% of the cases, and atrophy and stromal induction in 66.6% and 55.5%, respectively.

Androgen receptor (AR) blockade evasion is the most common form of therapeutic resistance in PCa and represents an example of tumor adaptation that develops uncontrolled castration-resistant clones [26]. Interestingly, a specific pattern of DNA hypermethylation predicts the development of castration-resistant cancer [27]. All the cases in the present series showed positive immunostaining for AR in the primary tumor but up to 29.6% of them achieved in the last contact a castration-resistant status.

ERG overexpression promotes the development of androgen independent PCa clones disrupting the androgen receptor [28]. In the clinical practice, the protein tested is generated by the *TMPRSS2-ERG* fusion genes being positive in a significant percentage of PCa [29]. This series of OPCa expressed this protein in the primary tumor of more than 40% of the cases.

The importance of neurogenesis in cancer has been very recently reviewed [30]. Axonogenesis identified with PGP 9.5 immunostaining is a predictor of biochemical relapse in the univariate test of PCa and shows a positive correlation with LN status in a series of 640 patients analyzed with computerized digital image analysis [31]. In the present series, up to 74% of the cases showed evident positive intratumor PGP 9.5 staining.

Several studies have confirmed the positive correlation between Ki-67 high index and bad prognosis in prostate cancer [26,32,33,34]. This finding is especially interesting in localized prostate cancer, where it is associated with poor disease-specific survival, biochemical failure-free survival, and other parameters of bad prognosis [34]. Also, a high Ki67 index is associated with increased risk of metastases [33] and a bad prognosticator in prostate adenocarcinomas with hormonal blockade [26].

True neuroendocrine differentiation (NED) is rare (0.5–2%) in PCa before the androgen deprivation therapy is administered but it is significantly increased thereafter (17–30%) [35]. Several types of NED are associated with a highly aggressive form of PCa, with rapid progression and metastases mainly to visceral organs [36]. After Epstein et al.’s classification [37], it seems that usual prostate adenocarcinomas with focal NED are quite common in the pre-androgen blockade period. Interestingly, this finding does not worsen the prognosis of patients [38].

The L1 cellular adhesion molecule (L1CAM) has been implicated in the development of metastases in a wide range of malignant tumors [38]. L1CAM expression has been rarely analyzed in prostate cancer, where it has been associated with androgen-insensitive prostate cancer and high metastatic potential [39]. Since inactivating L1CAM in prostate cancer inhibits metastases [39], this strategy could be an additional potential targeted therapy to be applied in tumors with known aggressive potential. However, the negative immunostaining in the present series suggest that this protein is not an actionable target in OPCa.

To our knowledge, this is the first study looking specifically at pathologic characteristics of OPCa patients. Here, we have found that routine pathological parameters like inflammation, atrophy, Ki-67 index, Gleason index, pathologic staging, and PGP 9.5, chromogranin A, and synaptophysin immunostaining may be relevant in order to achieve better local control rates and to detect castrate resistant disease.

To date, the decision to administrate local ablative treatments in OPCa patients is based only on the clinical characteristics of the patients. Research is being conducted nowadays looking at molecular biomarkers that could help to predict outcomes. Preliminary data suggest that tissue miRNAs may drive metastatic competence by adaptative communication processes between cancer cells and their microenvironment [40]. In this regard, our group has started to investigate the role of the transcriptional co-activator peroxisome proliferator-activated receptor gamma co-activator 1α (PGC1α) in these patients and its association with disease progression.

## 5. Conclusions

OPCa is a category of prostate carcinomas with increasing interest in clinics since the application of local MDT has proven to be very useful in recent years. The initial identification of prostate cancer patients belonging to this category is difficult because of the lack of specific clinical, histological, and immunohistochemical markers. In this study, we have seen that some histological (inflammation and atrophy) and immunohistochemical (Ki-67, PGP 9.5, chromogranin A, and synaptophysin) data are associated with a successful local control of disease and non-organ confined disease, thus helping to make therapeutic decisions. However, further studies are needed to define more precise prognostic identifiers in these patients.

## Figures and Tables

**Table 1 jpm-10-00265-t001:** Clinical data of 27 oligometastatic prostate adenocarcinomas.

Clinical Parameters	*n*	%
Patient age at MDT (mean, range)	67 (56–78)	
Initial treatment		
RP	7	25.9
RP + Salvage RT	20	74.1
PSA before first MDT (ng/mL) (mean, range)	3.41 (1.03–12.28	
PSA doubling time in months (mean, range)	4.5 (1–21.5)	
Follow-up (months) (mean, range)	54 (26–70)	
First oligometastases location		
Nodes	22	81.5
Bones	5	18.5
Total number of MDTs		
One MDT	17	63
Multiple MDTs	10	37

MDT: Metastases directed therapy, RP: Radical prostatectomy, RT: Radiotherapy.

**Table 2 jpm-10-00265-t002:** Pathological findings in 27 oligometastatic prostate adenocarcinomas.

	*n*	%
pT		
Organ confined (pT2a/b/c)	9	33.3
Non-organ confined (pT3a/b)	18	66.6
Histology		
Acinar	25	92.5
Ductal	2	7.5
Inflammatory infiltrates		
Mild	8	29.6
Moderate	10	37.1
Severe	9	33.3
Atrophy	18	66.6
Stromal reaction	15	55.5
Gleason index		
3 + 3	3	11.1
4 + 3	9	33.3
3 + 4	3	11.1
3 + 5, 4 + 4, 4 + 5, 5 + 4	12	44.4
Prognostic grouping (ISUP 2014)		
1 + 2	12	44.4
3 + 4 + 5	15	55.6
Androgen receptor	27	100
ERG	11	40.7
PGP 9.5	20	74
Ki-67 (cut-off 5%)		
≤5	23	85.2
>5	4	14.8
Chromogranin A	8	29.6
Synaptophysin	5	18.5
L1CAM	0	0

**Table 3 jpm-10-00265-t003:** Dependent parameters for successful local control, non-organ confined staging, and castration resistance status in 27 oligometastatic prostate adenocarcinomas (Fisher exact test).

		*p* Value
Successful local control	Low inflammation	0.0003
	No atrophy	0.0008
	No stromal reaction	0.0103
	Organ-confined tumor	0.002
	Gleason index 3 + 3	0.00001
	Gleason index (first pattern 3)	0.0214
	Ki67 < 5%	0.00001
	Ki67 ≤ 5%	0.0003
	ERG negative (IHC)	0.0047
	PGP 9.5 negative (IHC)	0.0001
	Chromogranin A negative (IHC)	0.0001
	Synaptophysin negative (IHC)	0.00001
Non-organ confined tumor	High inflammation	0.028
	Ki67 > 5%	0.00006
	Ki67 ≥ 5%	0.0281
	Chromogranin A positive (IHC)	0.01
	Synaptophysin positive (IHC)	0.002
Castration resistant tumor	High inflammation	0.006
	Atrophy	0.006
	Gleason index other than 3+3	0.00001
	Ki67 > 5%	0.0001
	Ki67 ≥ 5%	0.006
	PGP 9.5 positive (IHC)	0.0024
	Chromogranin A positive (IHC)	0.0024
	Synaptophysin positive (IHC)	0.05

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
