# Peer review of "Oligometastatic Prostate Adenocarcinoma. Clinical-Pathologic Study of a Histologically Under-Recognized Prostate Cancer"

_jpm, 2020, doi:10.3390/jpm10040265_

Round 1

Reviewer 1 Report

It is a kinds of non-analytic study with pretty small-sample size (N=27).  And the study choose some common immunohistory staining proteins such as AR, ERG, Ki-67 to characterize OPCs on top of description of clinic-pathological features.

My major concerns are listed as follows:

1) There is the major different way to present statistic data between table 1 and table 2.

2) I am pretty confused about table 3, which does not clarify the comparison groups

3) The author should give more explanation of the criteria of choosing AR, Ki-67 and ERG as characterization markers.

Author Response

Thank you for your comments and suggestions

  • Statistics have been presented in Table 1 and 2 in the same way.
  • A new more informative heading for Table 3 has been added
  • AR, Ki-67 and EGR antibodies, like the other ones are antibodies of biological aggressiveness and, a sentence in this sense has been added in M&M (highlighted in red)

Reviewer 2 Report

Prostate cancer is the most common neoplasm and a leading cause of mortality in the male of Western countries. Oligometastatic prostate cancer (OPCa) is a tumor category strictly defined by clinical-radiological criteria. Claudia Manini et al., found that some histological and immunohistochemical markers were associated with successful local control of disease and non-organ confined disease. Mild inflammation, organ confined disease, Ki-67 and Gleason index were significantly not related with OPCa. Authors hoped further studies were needed to define more prognostic markers related to OPCa.

The study’s conclusion showed that clinical and pathological characteristics, recurrence and oncological outcomes in 27 patients. Authors also discussed and compared the previous papers and their data in detail. OPCa is an interesting and important topic and concept. The current study is interesting, but there are some major concerns that need to be addressed.

Major concerns:

  1. This study was interesting, but there is not enough figure panels to prove the evidence shown in the paper. Authors should show the Ki-67 staining, and PET/CT scan to let the readers know how they define the “strong or weak” of these markers. The table was a suitable way to show the number of these markers, how was the strength of some markers.
  2. There were multiple standards to define the OPCa (Tabata K et al., 2012; Ahmed KA et al., 2013; Ost P et al., 2016; Decaestecker K et al., 2014; Schick U et al., 2013), how authors define OPCa in this paper was not detail.
  3. The clinical detailed data should be in supplemental tables, like PSA levels, inflammatory infiltrates, pT, Histology and Gleason index of each patient. Thus reads can know how strength of each markers in one patients and can compared with other papers to easily cite this paper.
  4. The figure legends were too simple. Please indicate in all experiments and legends the number of statistical methods, and data measuring standards.

Minor concerns:

The Numbers shown in Table 2 were “comma” inside, but in Table3 were “dot” inside, authors should make a consistent.

Author Response

Thank you for your comments and suggestions

  • Ki-67 is marked with a cut-off, as usual with this marker in routine practice. In most practical routine samples, a immunostaining of more than 5% of the nuclei in the proliferating cells is considered a frontier of aggressiveness. A picture of this finding does not add anything interesting to the paper since, although speaking of aggressiveness, any cut-off analyzed was a specific characteristic of oligometastatic prostate adenocarcinoma. The same happens with PET/CT scans; there is no feature specifically linked to this prostate cancer. So we conclude that pictures do no add any relevant information deserving a space in this manuscript.
  • As stated in the text (reference #9) we have followed the inclusion criteria proposed by Tosoian, J.J.; Gorin, M.A.; Ross, A.E.; Pienta, K.J.; Tran, P.T.; Schaeffer, E.M. Oligometastatic prostate cancer: definitions, clinical outcomes, and treatment considerations. Rev. Urol. 2017, 14, 15-25.
  • Relevant clinical data (initial treatment, PSA before the first metastasis directed treatment, PSA doubling time, oligometastases location, etc) have been included in Table 1.
  • Table legends and tables themselves have been completed.
  • “Commas” have been modified in Table 2 for consistency

Reviewer 3 Report

The manuscript by Manini et al entitled “Oligometastatic prostate adenocarcinoma. Clinical-pathologic study of a histologically under-recognized prostate cancer” is an interesting histological and immunohistochemical study about patients-derived oligometastatic prostate adenocarcinomas. There have been very few articles on this topic, making any new contribution to the field interesting. Overall, the manuscript is well written, with several tables clearly summarizing the text. I only have few minor comments:

- The tables title (1,2 and 3) should be placed above the same table

- Some typographical errors are present. For instance, see line 96 (2ng/mL), line 165, references list and check the whole text

Author Response

Thank you for your comments and suggestions

  • Table titles have been replaced on the top of tables
  • Text has been checked throughout for typographical errors.